# NAD^+^ Metabolism and Interventions in Premature Renal Aging and Chronic Kidney Disease

**DOI:** 10.3390/cells12010021

**Published:** 2022-12-21

**Authors:** Lucie Chanvillard, Alessandra Tammaro, Vincenzo Sorrentino

**Affiliations:** 1Nestlé Institute of Health Sciences, Nestlé Research, 1015 Lausanne, Switzerland; 2School of Life Sciences, Ecole Polytechnique Fédérale de Lausanne (EPFL), 1015 Lausanne, Switzerland; 3Department of Pathology, Amsterdam UMC location University of Amsterdam, 1105AZ Amsterdam, The Netherlands; 4Amsterdam Infection & Immunity, 1105AZ Amsterdam, The Netherlands; 5Healthy Longevity Translational Research Programme, Yong Loo Lin School of Medicine, National University of Singapore, Singapore 119228, Singapore; 6Department of Biochemistry, Yong Loo Lin School of Medicine, National University of Singapore, Singapore 117596, Singapore

**Keywords:** NAD^+^, NAD^+^ precursors, age-related diseases, premature renal aging, chronic kidney disease, kidney, tubular epithelial cells metabolism, mouse models, clinical trials

## Abstract

Premature aging causes morphological and functional changes in the kidney, leading to chronic kidney disease (CKD). CKD is a global public health issue with far-reaching consequences, including cardio-vascular complications, increased frailty, shortened lifespan and a heightened risk of kidney failure. Dialysis or transplantation are lifesaving therapies, but they can also be debilitating. Currently, no cure is available for CKD, despite ongoing efforts to identify clinical biomarkers of premature renal aging and molecular pathways of disease progression. Kidney proximal tubular epithelial cells (PTECs) have high energy demand, and disruption of their energy homeostasis has been linked to the progression of kidney disease. Consequently, metabolic reprogramming of PTECs is gaining interest as a therapeutic tool. Preclinical and clinical evidence is emerging that NAD^+^ homeostasis, crucial for PTECs’ oxidative metabolism, is impaired in CKD, and administration of dietary NAD^+^ precursors could have a prophylactic role against age-related kidney disease. This review describes the biology of NAD^+^ in the kidney, including its precursors and cellular roles, and discusses the importance of NAD^+^ homeostasis for renal health. Furthermore, we provide a comprehensive summary of preclinical and clinical studies aimed at increasing NAD^+^ levels in premature renal aging and CKD.

## 1. Introduction

The process of aging has been associated with a range of structural and functional changes in the kidney and a decreased ability to recover from a kidney injury, contributing to long-term renal outcomes [1]: over 60% of people aged more than 80 are diagnosed with a form of chronic kidney disease (CKD) [2]. Overall, CKD is highly prevalent; it is estimated to affect >10% of the adult population worldwide, or more than 800 million people in 2022 [3]. It is therefore a crucial health problem due to an increasingly aging population.

CKD is defined as persistent abnormalities of kidney function or structure, present for at least 3 months, with implications on health [4]. Patients progress from early-stage symptoms to kidney failure, which requires dialysis or a kidney transplant [5]. The Kidney Disease Improving Global Outcomes (KDIGO) guidelines for the evaluation and management of CKD proposed in 2012 a new classification of CKD. This is based on cause and a combination of both the degree of kidney function, measured by estimated glomerular filtration rate (eGFR), and the degree of kidney damage, measured by albuminuria, and therefore called CGA staging [6]. The prognosis for CKD progression is higher when eGFR, i.e., the estimated blood volume passing through the glomeruli, is inferior to 60 mL/min/1.73 m^2^ and is accompanied by albuminuria, i.e., the presence of albumin in the urine, greater than 30 mg per g of creatinine [6]. The main histopathological feature of CKD is the progression of renal interstitial fibrosis, triggered by multiple and different stimuli, leading to organ dysfunction [7]. Interstitial fibrosis/tubular atrophy (IFTA) severity is incorporated into most pathology scoring systems and is used to predict renal outcomes in nephrosclerosis [8] and diabetic nephropathy [9].

CKD leads to premature aging of other organs, due to the central role of the kidney in body homeostasis, regulating blood volume and plasma osmolarity. It is indeed associated with many complications, such as cardiovascular disease (CVD), bone mineral disorder (BMD), metabolic acidosis, and anemia, all of which significantly shorten lifespan [4]. Identifying and targeting biological mechanisms that may reduce the risk of premature kidney aging could then have an overall impact on health status and promote healthy aging.

At the moment, there is no effective cure for patients with CKD until they reach a stage requiring renal replacement therapy. Current drug therapies aim at retarding the disease progression through the blocking of either the renin-angiotensin system driving the vasoconstriction in hypertensive nephropathies or of the glucose reabsorption via sodium-glucose transporter 2 (SGLT2) inhibitors to lower the glycemia in Diabetic Kidney Disease (DKD) [10]. Nevertheless, nutritional intervention approaches appear also to be beneficial in CKD patients, preventing further damage from macro- and micronutrients. Both the European Society for Clinical Nutrition and Metabolism (ESPEN) and the US National Kidney Foundation (NKF) recommend lower protein intake for non-dialyzed patients (~0.6 g/kg body weight per day) but higher protein intake in case of hemodialysis or peritoneal dialysis (~1.2 g/kg body weight per day) [11,12]. In addition, nutritional guidelines include low phosphate, potassium, and sodium intake and supplementation with vitamins B9, B12, C, and D in case of deficiency. Moreover, plant-based diets and bioactive nutrients have been shown to alleviate some features of CKD, including gut dysbiosis, inflammation, and mitochondrial dysfunction, but this is not sufficient for effective clinical management of the disease [13]. This warrants the need for further studies aimed at identifying effective CKD interventions targeting the early stages of premature renal aging, before the development of fibrosis.

Accumulating evidence suggests that the central locus of premature renal aging is the renal proximal tubule (PT), the first segment of the nephron after the glomerulus, where reabsorption begins [14]. PT accounts for ~90% of the cortical mass of the kidney and 50% of the volume of the normal kidney, and markers of tubular injury can be detected along with a GFR decline during both acute kidney injury (AKI) and CKD [15,16]. The primary function of proximal tubular epithelial cells (PTECs), the main cell population forming the PT, is to reabsorb sodium and water and transport solutes against their gradients. To produce the adenosine triphosphate (ATP) required for active transporters, PTECs use 10% of the total oxygen consumption of the body [17]. Thus, the PT is rich in mitochondria and relies heavily on fatty acid β-oxidation (FAO) as fuel for energy [18]. The high energy dependence of PTECs renders them vulnerable to oxidative and metabolic stress, which disrupt mitochondrial function in premature renal aging [19]. Available data suggest that these cells can rewire their metabolism after being exposed to cellular stress, resulting in the lipid accumulation characteristic of kidney disease [20,21].

In particular, recent preclinical and clinical evidence points to changes in Nicotinamide Adenine Dinucleotide (NAD^+^) metabolism, primarily in the PT, linked with renal aging and CKD staging [22,23]. NAD^+^ and its reduced form NADH, by regulating redox reactions and allowing the production of ATP, are crucial for energy metabolism and FAO in PTECs. In addition, by being the substrate of non-redox NAD^+^-consuming enzymes such as sirtuins and poly(ADP-ribose) polymerases (PARPs), NAD^+^ is also involved in several key molecular mechanisms for cellular homeostasis [24]. Therefore, NAD^+^ supplementation has been shown to delay the progression of several age-related diseases and improve lifespan and healthspan in several animal models [25].

In this review, we present the biology of NAD^+^ in the kidney, including its cellular roles and precursors. Next, we discuss the importance of NAD^+^ homeostasis for renal health and propose mechanistic links between NAD^+^ depletion, CKD, and aging. Finally, we present preclinical and clinical studies aimed at increasing NAD^+^ levels in premature renal aging and CKD, and future perspectives for this field.

## 2. NAD^+^ Biology in the Kidney

NAD^+^ is present in the kidney at concentrations ranging from 0.3 to 1 mmol/kg of tissue, which is comparable to concentrations found in liver and muscle [22,26,27]. In this section, we detail the NAD^+^ biosynthetic pathways from different precursors in PTECs, and describe its cellular roles in renal tissue, before explaining the alterations in NAD^+^ homeostasis occurring during premature renal aging and CKD.

### 2.1. NAD^+^ Homeostasis and Precursors

Mammals can produce NAD^+^ from different dietary precursors, which include the essential amino acid L-tryptophan (L-Trp), different forms of vitamin B_3_, and the NAD^+^ breakdown products that contain a pyridine ring. To date, there are at least five different pathways for NAD^+^ production that are grouped into (i) the *de novo* biosynthesis pathway from L-Trp; (ii) the Preiss-Handler pathway fueled by nicotinic acid (NA); and (iii) the amidated salvage pathways which include the salvage of nicotinamide (NAM) to nicotinamide mononucleotide (NMN), the conversion of nicotinamide riboside (NR) to NMN as well, and the conversion of the reduced form of NR, dihydronicotinamide riboside (NRH), to dihydronicotinamide mononucleotide (NMNH) (Figure 1). The kidney expresses all of the NAD^+^-synthesizing enzymes, most of them abundantly [24], and is therefore able to take advantage of all five pathways. Quantitative analysis of in vivo NAD^+^ fluxes with labeled precursors confirmed that the kidney is the only other organ, after the liver, that can produce and excrete NAD^+^, primarily as NAM [28]. However, the kidney generates about 5% of the total circulating NAM, so renal NAD^+^ production is unlikely to be a critical driver of systemic NAD^+^ needs. In addition, kidney and liver, as well as the other tissues, take up circulating NAM or other NAD^+^ precursors and “salvage” it as NAD^+^.

Even though NAD^+^-synthesizing enzymes are present in both the tubulointerstitial and glomerular compartments of the kidney [29], the focus here is on the metabolism of PTECs, given their high metabolic demand. Indeed, prior to use it, PTECs take up NAD^+^ and its precursors at their brush borders, through the sodium-monocarboxylate transporter SLC5A8 (SMCT) or SLC22A13 for NA and through unidentified transporters for NAM [30,31]. NMN as well as NR and its reduced form NRH are imported by equilibrative nucleoside transporters (ENTs) [27,32]. L-Trp is taken up by neutral amino acid transporters [33] (Figure 2). The specific biosynthetic pathways of NAD^+^ metabolism, its excretion and its compartmentalization are described below.

#### 2.1.1. De Novo Pathway

The nine-step de novo pathway allows the synthesis of kynurenine from L-Trp, which can be processed into various signaling molecules or give rise to amino-β-carboxymuconate-ε-semialdehyde (ACMS). ACMS can in turn be processed by ACMS decarboxylase (ACMSD) or undergo spontaneous cyclization to form quinolinic acid [34]. QA can then only be converted by the enzyme quinolate phosphoribosyl transferase (QPRT) to nicotinic acid mononucleotide (NAMN), which then feeds into the Preiss-Handler pathway.

The kidney is, along with the liver, one of two organs capable of converting L-Trp to NAD^+^, and both tissues appear to prefer this precursor to NA based on quantitative analysis of in vivo NAD^+^ flux by Liu et al. [28]. Loss-of-function mutations in two de novo enzymes, 3-hydroxyanthranilate 3,4-dixoygenase (*HAAO*) and kynureninase (*KYNU*), are associated with multiple congenital malformations, including major renal abnormalities, resulting in a reduction in circulating NAD^+^ levels in patients [35].

#### 2.1.2. Preiss-Handler Pathway

The Preiss-Handler pathway, or deamidated salvage pathway, consists of three steps [36]: nicotinic acid phosphoribosyltransferase (NAPRT) produces NAMN from the dietary NA or utilizes NAMN from the L-Trp processing, and nicotinamide mononucleotide adenylyltransferases (NMNATs) catalyze the addition of the adenylyl moiety from ATP to form nicotinic acid adenine dinucleotide (NAAD), which is finally amidated to NAD^+^ by the NAD^+^ synthetase (NADS) [37].

In mice, NA has been shown to increase renal NAD^+^ levels, but less efficiently than NAM [38]. Indeed, the deamidated route accounts for a minority of the NAD^+^ synthesized in mice kidneys, as evidenced by the low activity of NADS compared to NMNATs, which are the last-step enzymes in both routes [39]. While NA and its derivatives have already been used clinically for CKD, primarily because of its phosphate-lowering properties (see Section 3.2) the actual physiological importance of the Preiss-Handler pathway in human kidney NAD^+^ homeostasis remains to be assessed.

#### 2.1.3. Amidated Salvage Pathways

The amidated salvage pathways include recycling of NAM (sometimes referred to exclusively as the salvage pathway [40]), that is one of the breakdown products of the NAD^+^-consuming enzymes, and the phosphorylation of dietary NAD^+^ precursor NR and its reduced form NRH. The enzymes that catalyze these three reactions are NAMPT [41], NR kinases (NRKs) [42] and adenosine kinase (AK) [27], respectively, and the product is NMN/NMNH, which is then converted to NAD^+^/NADH by NMNATs.

NAMPT is the rate-limiting enzyme for the recycling of NAM to NAD^+^ [43] and is controlled by the circadian clock machinery, as are the intracellular NAD^+^ levels [44,45]. Homozygous *Nampt* KO mice are embryonically lethal, and heterozygous animals have lower NAD^+^ levels in tissues that do not generate de novo NAD^+^ [46], but also in the liver, a tissue with an active de novo pathway [47], suggesting that NAM salvaging is the major pathway for NAD^+^ production in mammals.

NRK1 is essential for the maintenance of energy homeostasis and catalyzes the rate-limiting step for the efficient utilization of exogenous NR and NMN [48]. Indeed, it seems that NMN requires to be dephosphorylated by CD73 before entering the cells as NR where it can be phosphorylated back to NMN [49], even though a specific NMN transporter highly expressed in the small intestine has been recently discovered [50]. In addition, NRKs can phosphorylate nicotinic acid riboside (NAR), the deamidated form of NR, to generate NAMN that can thus enter the Preiss-Handler pathway [51]. Both NR and NAR can be hydrolyzed by purine nucleoside phosphorylase (PNP) to NAM and NA, respectively [52].

Recently, the reduced form of NR, NRH, has gained interest as a very potent precursor of NAD^+^, especially in the kidney [27,53]. NMNH has also been shown to increase NAD^+^ levels after a dephosphorylation step to NRH prior to its incorporation into the intracellular space [54]. A final reduced precursor, dihydronicotinic acid riboside (NARH) can act synergistically with nicotinamide riboside (NR) to increase NAD^+^ through their conversion to NRH [26]. The greater effectiveness of NRH compared to NR as an NAD^+^ booster could be based on both the higher enzymatic phosphorylation rate of AK compared to NRK1, and the fact that NRH is not or less degraded to NAM in plasma, which is the case for NR [27].

#### 2.1.4. Phosphorylation to NADP^+^

Two other fates of the NAD^+^ molecule, if it is not consumed and recycled or reduced, are to be noted: its phosphorylation and its excretion. The first, which represents about 10% of total NAD(H) [28], involves NAD^+^ kinase (NADK) to produce NADP^+^ [55]. The transhydrogenase activity of different enzymes leads to a rapid conversion to NADPH, both in the cytosol and in the mitochondria, favoring the reduced form of the NADP^+^/NADPH couple as opposed to NAD^+^/NADH [56,57]. In turn, NADP^+^ can be converted back to NAD^+^ by NADP^+^ phosphatases [58].

#### 2.1.5. Excretion

The last metabolic route of NAD^+^ homeostasis is the excretion of its metabolites from the body via the urine. The enzyme nicotinamide N-methyltransferase (NNMT) catalyzes the methylation of NAM to form methylated NAM (MeNAM) [59] that can be further oxidized to N-methyl-2-pyridone-5-carboxamide (N-Me-2PY) and N-methyl-4-pyridone-3-carboxamide (N-Me-4PY) [60]. In mammals, NNMT expression and activity appear to increase during aging and age-associated diseases without a clear explanation of its relevance in the healthy state or during aging pathogenesis [22,61,62,63]. In contrast, interestingly, the NNMT homolog in the short-lived nematode *Caenorhabditis elegans*, ANMT-1, extends lifespan. This is due to reactive oxygen species (ROS) generated by an aldehyde oxidase using MeNAM as a substrate, resulting in a mitohormesis signal [64].

#### 2.1.6. Compartmentalization

NAD^+^ can move freely between the intracellular compartments of cytosol, nucleus, and mitochondria (Figure 2). The recent discovery of a mammalian mitochondrial NAD^+^ transporter [65] has overturned the long-held theory that mitochondria rely only on imported NMN, salvaged by the specific enzyme NMNAT3 [66,67]. However, mitochondrial NAD^+^ levels and half-life differ from the nucleo-cytosolic pool and appear to be independently regulated [68]. While in most tissues and cells NAD^+^ appears to be more present and stable in mitochondria [69], in hepatocytes, the majority of NAD^+^ is cytosolic [70] which is also consistent with the subcellular localization of the de novo pathway enzymes in liver [24]. Given that the de novo pathway has also been described in the kidney, the renal NAD^+^ pool may be present in both mitochondria and cytosol, yet to be determined. Finally, the subcellular distribution of NAD^+^ seems to vary according to the metabolic needs of the cells by locally activating specific NAD^+^-consuming enzymes and not to disrupt the activity of others [71]. Detailed description of the subcellular distribution of NAD^+^ and its biosynthesis enzymes has been reviewed by Katsyuba et al. [24].

### 2.2. NAD^+^ Cellular Roles

#### 2.2.1. Redox Co-Factor

NAD^+^ is an essential cellular metabolite for every cell. First described in 1906 as a crucial component for yeast fermentation [72], it took 30 years to understand the ability of this thermostable cofactor to accept a hybrid ion and transfer it to molecules [73].

In both the cytosol and mitochondria, NAD^+^ acts as an electron acceptor in several catabolic reactions, including glycolysis, the tricarboxylic acid (TCA) cycle, FAO and alcohol metabolism [74]. NAD^+^ is regenerated from its reduced form NADH by lactate dehydrogenase (LDH), which catalyzes the reduction of pyruvate to lactate, primarily under anaerobic conditions, or by mitochondrial oxidative phosphorylation (OXPHOS) complex I which provides electrons through the electron transport chain (ETC) to ultimately synthesize ATP in the presence of oxygen. Under aerobic conditions, to allow OXPHOS reactions to occur at the mitochondrial inner membrane, the NADH molecules produced in the cytosol transfer into the mitochondria with the malate/aspartate shuttle.

The NADP^+^/NADPH pair, on the other hand, is mainly required for anabolic reactions, such as the pentose phosphate pathway and the synthesis of fatty acids and steroids [75]. In addition, NADPH plays a role in oxidative mechanisms in renal diseases, both by being used by NADPH oxidase for the respiratory burst, i.e., the rapid production of ROS for immunological defense, but also for the reduction of glutathione to detoxify oxidative stress [76].

#### 2.2.2. Enzymatic Substrate

In the 2000s, NAD^+^ regained much attention by being described as a substrate for metabolic and aging coordinating enzymes, the sirtuins [77]. Seven mammalian sirtuins have been identified, orthologs of the yeast silent information regulator Sir2 which promotes longevity through transcriptional hyper-silencing [78]. Sirtuins have been proposed as the key mediators of caloric restriction-induced life extension [79,80]. Sirtuins localize to distinct subcellular compartments (both nucleus and cytosol for SIRT1, nucleus for SIRT6, nucleolus for SIRT7, mitochondria for SIRT3, SIRT4, and SIRT5, and cytosol for SIRT2 and SIRT5) and differ in their enzymatic activities and targets. Sirtuins can deacetylate lysine residues of various proteins, including histones H1, H3, and H4, to condense chromatin and induce gene silencing, as well as transcription factors such as peroxisome proliferator-activated receptor-gamma coactivator (PGC)-1alpha, FoxO1, nuclear factor-kappa B (NF-κB), and p53 to modulate their activity and thus regulate metabolism. In addition, some sirtuins can deacylate other post-translational modifications (succinylation, malonylation, and fatty acid acylation). These reactions result in the production of the deacetylated substrate, NAM and 2-O-acyl-ADP-ribose as end products. Finally, SIRT4 and SIRT6 have mono-ADP-ribosyl transferase activity, also releasing NAM [81,82]. Interestingly, NAM non-competitively binds and thus feedback-inhibits sirtuins at least in the simple model *Saccharomyces cerevisiae*, underscoring the importance of NAD^+^ metabolites fluxes in cells [83,84].

In the kidney tissue, the most studied sirtuins are SIRT1 and SIRT3, with a recent interest in SIRT6. SIRT1 and SIRT3 are the only two sirtuins whose Km value can exceed the physiological range of NAD^+^ in the body (~300–700 μM), meaning that their activity is highly dependent on NAD^+^ availability [24]. SIRT1 is a key factor in mitochondrial quality maintenance, by promoting mitochondrial biogenesis through PGC1α deacetylation and the removal of defective mitochondria by mitophagy [85]. In the kidney, SIRT1 is involved in important signaling pathways of metabolic homeostasis and stress responses such as mitochondrial biogenesis, but also apoptosis of PTECs, autophagy and inflammation, and ultimately plays a role in renal fibrosis [86]. Homozygous *Sirt1* knockout (KO) mice exhibit severe developmental defects [87] whereas heterozygous *Sirt1* KO mice show more fibrosis in the model of unilateral ureteral obstruction (UUO)-induced CKD [88]. For diabetic nephropathy, PT-specific *Sirt1* transgenic mice showed prevention of albuminuria and glomerular changes in streptozotocin (STZ)-treated and *db*/*db* mice while PT-specific *Sirt1* KO mice showed worsening of the same parameters in STZ-treated mice [89]. The mitochondrial SIRT3 regulates lipid metabolism and the antioxidant system [90,91], and *Sirt3* KO mice exhibit hyper-acetylated mitochondrial proteins in PTECs and severe renal fibrosis [92]. Finally, SIRT6 plays an important role against renal fibrosis. Its upregulation after UUO or ischemia/reperfusion injury (IRI) has been mechanistically explained in PTECs by its role in modulating the Wnt/β-catenin signaling pathway [93]. PT-specific *Sirt6* KO mice show tubular basement membrane (TBM) thickening and collagen deposition, associated with the upregulation of the profibrogenic gene *TIMP-1* [94]. Importantly, the phenotype is similar to that of *Nampt* KO mice, suggesting a convergent mechanism between loss of SIRT6 and dysfunction of NAD^+^ biosynthesis [94]. Several natural and synthetic sirtuin-activating compounds (STACs) have shown beneficial effects in preclinical models of kidney disease [95]. Resveratrol, the most widely used natural STAC, enhanced SIRT1 activity and ameliorated tubular fibrosis, in the 5/6 nephrectomy [96] and the UUO models [97]. The mechanism in PTECs was found to be via deacetylation of Smad 3 [97] or Smad 4 [98], important transcription factors of the transforming growth factor-beta (TGFβ) pathway. Similarly, honokiol, a STAC activating SIRT3, protected against UUO-induced tubular fibrosis through the regulation of the NF-κB/TGF-β1/Smad signaling pathway and mitochondrial dynamics [99].

NAD^+^ is also a substrate for poly(ADP-ribose) polymerases (PARPs) that transfer its ADPr moiety to nuclear proteins, thereby releasing NAM. The target proteins end up with long chains of ADPr homopolymers; this post-translational modification is called PARylation and is reversible. The PARP family, which has 17 members in humans and 16 in mice, plays a key role in the maintenance of genomic stability [100]. Specifically, PARylation of histones and other proteins, including PARP itself, enables the recruitment and activation of DNA repair enzymes at the damaged site [101]. PARP 1 and PARP 2 account for about 90% and 10% of the NAD^+^ used by the PARP family, respectively [102], which is about one-third of the total NAD^+^ turnover under basal conditions but can be up to 60% under overt DNA damage [28], depleting intracellular NAD^+^ levels. The low Km of PARPs for NAD^+^ suggests that their enzymatic activity may not be limited when NAD^+^ levels decrease [24], whereas it may become limiting for sirtuins, as observed after genotoxic stress or during aging and which partly explains the altered mitochondrial metabolism resulting from these conditions [103]. PARP 1/2 inhibition could therefore alleviate diseases with PARP overactivation by increasing the availability of NAD^+^ for sirtuins [104]. Consistently, *Parp1* KO mice are more resistant to functional and histological damages following ischemic AKI [105], UUO [106], and STZ-induced diabetic nephropathy [107] than their wildtype littermates. The mechanism converges towards a decrease neutrophil infiltration and reduction in expression of proinflammatory proteins [105,106].

Beyond sirtuins and PARPs, NAD^+^ is also consumed by CD38, CD157 and SARM1. The cell surface protein CD38 and its homologue CD157 are multifunctional enzymes and cell receptors important for the immune system. First identified on T cells and on bone marrow stromal cells, CD38 and CD157, respectively, are now known to be expressed in several tissues, including the kidney [108,109]. Both are primarily NAD^+^ glycohydrolases (with a low Km [24]), have an inefficient ADP-ribosyl cyclase activity, and can also mediate base-exchange reactions to produce nicotinic acid adenine dinucleotide (phosphate) (NAAD(P)) and NAM from NAD(P) and NA, or NAR and NAM from NR and NA, respectively for CD38 and CD157 [108,110]. The three end products APDr, cyclic ADPr (cADPr) and NAAD(P) are second messengers that mobilize intracellular calcium [111]. CD38 and CD157 are mainly ectoenzymes catabolizing NAD^+^ in the extracellular environment, but CD38 may also have its catalytic site facing the intracellular environment [112]. Moreover, CD38 has also been identified as an ecto-NMNase [113,114] and CD157 as an ecto-NRase [108,115], degrading NAD^+^ precursors before their entry into cells. The enzymatic activities of CD38 and CD157 help explain the decrease in NAD^+^ levels associated with their upregulated expression in aging tissues [61,113,114]. Pharmacological inhibition of CD38 in old mice increased NAD^+^ levels in several tissues, including skeletal muscle, liver, and spleen, but they were not measured in the kidney [116]. Finally, SARM1, primarily expressed in neurons, also possesses NAD(P)^+^ glycohydrolase, ADP-ribosyl cyclase, and base exchange activities [117,118]. NAD^+^ degradation by SARM1 has been linked to axonal degeneration [119]. Although SARM1 appears to be expressed in the kidney, its functionality has not yet been explored [120].

### 2.3. NAD^+^ Alterations in the Diseased Kidney

The first link between NAD^+^ levels and health was established in 1937, when Pellagra disease was found to be caused by dietary niacin deficiency [121]. Since then, there have been numerous reports describing a decline in NAD^+^ levels in many pathological conditions but especially in the context of aging [122]. This decline has been associated with several established molecular hallmarks of aging including mitochondrial dysfunction, DNA damage, cellular senescence, and loss of proteostasis, and may be causally linked to several pathologies, including premature renal aging and kidney diseases [122]. Indeed, the kidneys of old mice have lower levels of NAD^+^, correlating with increased expression of fibrotic genes and impaired mitochondrial respiration [123], as well as decreased expression of SIRT1 linked to greater susceptibility to kidney injury [23]. In humans, the decrease of NAM and NMN circulating metabolites correlates with CKD staging [22]. Systemic loss of NAD^+^ levels is caused by (i) an imbalance in synthesis, due to altered expression of NAD^+^-homeostasis enzymes, and (ii) increased NAD^+^ consumption [124] (Figure 2).

#### 2.3.1. Impaired NAD^+^ Production

Since the study by Poyan-Mehr et al. revealed that the *de novo* NAD^+^ biosynthetic pathway is impaired in ischemic AKI, due to reduced expression and activity of QPRT [125] (Figure 1), the defective de novo pathway has been linked to AKI-to-CKD progression, a premature renal aging phenotype and staging of CKD in humans. The persistence of a high urinary quinolinate/tryptophan ratio (uQ/T), an indirect indicator of impaired QPRT activity that is an early marker of AKI, has been shown to predict progression to CKD in the clinic [126]. In addition, the expression of the enzymes kynurenine 3-monooxygenase (*KMO*), *KYNU*, *HAAO*, and *QPRT* is downregulated in a stage-dependent manner in human tubulointerstitial and glomerular compartments [29]. A genome-wide association study (GWAS) identified variations in the *KMO* gene associated with albuminuria [127], and a follow-up study confirmed that *KMO* expression is downregulated in human and mouse diabetic kidney glomeruli. Furthermore, knockdown of *Kmo* expression in zebrafish by morpholinos injection into fertilized eggs and KO of *Kmo* in mice each led to a spontaneous proteinuria phenotype and podocyte foot process effacement [128]. These results correlate with a decrease in *de novo* enzymes expression in preclinical models of CKD; mRNA expression of *Afmid*, *Kmo*, *Kynu*, *Haao* and *Qprt*, is downregulated in UUO- and POD-ATTAC-injured mice (model of progressive secondary tubulointerstitial lesions induced by high-range glomerular proteinuria) [29]. Protein expression of QPRT is also decreased in models of 5/6 nephrectomy and adenine-induced nephropathy [129]. Concurrently, in patients with CKD, the activity of the enzymes catalyzing the first step of the de novo pathway, the conversion of L-Trp to N-formylkynurenine, namely tryptophan 2,3-dioxygenase (TDO) and indoleamine 2,3-dioxygenase (IDO) is increased [29,130], as IDO is known to be sensitive to chronic inflammation, a feature of premature renal aging [131]. Increased TDO and IDO activity results in decreased circulating L-Trp levels and significant release of kynurenine metabolites such as kynurenine, kynurenic acid, and quinolinic acid [132,133,134,135], which are classified as protein-bound uremic toxins and linked to vascular complications [136]. In kidney transplant patients, for whom IRI is an inevitable consequence of transplantation, serum and urine kynurenine/tryptophan ratios are increased and correlate with graft rejection. Indeed, higher IDO immunostaining has been detected in biopsies from rejected kidneys, particularly in PTECs [137]. Importantly, it is also possible to improve NAD^+^ de novo biosynthesis, as Tran et al. showed that activation of PGC-1α can stimulate NAD^+^ de novo biosynthesis and metabolic flexibility in renal tubules, resulting in less fat accumulation and protection against worsening of kidney injury [138]. Conversely, NAD^+^ regulates PGC1α through its deacetylation by SIRT1.

NAMPT, the major enzyme of the amidated salvage pathway, is significantly decreased in multiple tissues during aging and metabolic disorders [139,140]. In the context of kidney disease, downregulation of NAMPT expression correlates with albuminuria and a fibrotic phenotype in STZ-treated diabetic mice [94] and has also been detected in the cisplatin-induced AKI rodent model [141,142]. Chronic inflammation mediated by inflammatory cytokines and oxidative stress is thought to be the reason why NAMPT expression is reduced during aging and age-related diseases [143]. However, surprisingly, opposing studies have revealed that glomerular cells (mesangial cells and podocytes) as well as PTECs increase NAMPT expression in diabetic nephropathy rat models, Otsuka Long-Evans Tokushima fatty (OLETF) rats [144] and STZ-treated rats [145,146]. In these studies, exogenous NAMPT has a proinflammatory effect on mesangial cells, explained by increased glucose uptake [147] and activation of NF-κB [145], while endogenous NAMPT protects PTECs from cell death in a proinflammatory environment [146]. This could be explained by the higher concentrations of exogenous/circulating NAMPT, which could also be released from fat and liver [146], and its insulin-mimicking effect further inducing glucotoxicity in the kidney cells of diabetic animals.

Additionally, *Nmnat1* and *Nrk1* are downregulated in the UUO model, whereas *Nmnat1*, *Nmnat3* and *Nrk1* are down-regulated in the POD-ATTAC model [29]. Boosting the salvage of NAM alleviates the STZ-induced diabetic nephropathy phenotype as the albuminuria of PT-specific *Nampt*-overexpressing transgenic mice is ameliorated [94].

Lastly, NNMT has been observed to be upregulated during CKD in humans and in the UUO mice model, resulting in decreased NAM salvage [22]. The final metabolites N-Me-2PY and N-Me-4PY are considered uremic toxins present at extremely high serum concentrations in CKD patients compared to healthy subjects [148].

#### 2.3.2. Accelerated NAD^+^ Consumption

During aging and under pathological conditions, including but not limited to those leading to CKD, PARPs and CD38 are increased in expression and turnover activity, leading to accelerated NAD^+^ consumption [122,149].

In rats subjected to renal IRI, *Parp1* expression was increased in damaged renal proximal tubules starting 6–12 h after reperfusion and maintained for several days [150]. After 10 days of UUO in mice, PARP1 expression and activation in the kidney were strongly increased by 6- and 13-fold, respectively [106]. The poly(ADP-ribosyl) protein content of the renal cortex was increased by 18% in STZ-treated diabetic mice compared with controls [107]. Clinically, PARP-1 expression in kidney transplant biopsies appears to correlate with cold ischemia time, acute tubular necrosis and renal function at biopsy [151,152].

CD38 NAD^+^ glycohydrolase is upregulated in both PTECs and glomeruli of Zucker diabetic fatty rat’s kidneys and is associated with a decrease in the NAD^+^/NADH ratio in the rodent renal cortex and in mitochondrial protein extracts isolated from the cortex [153,154]. In a lipopolysaccharide (LPS)-induced acute kidney injury model, CD38 expression was markedly increased on renal macrophages, resulting in their M1 polarization [155]. Currently, there is no clinical observation of increased CD38 expression or activity in the context of kidney disease yet; further studies will be needed to confirm the relevance of this NAD^+^ consumption pathway in the pathogenesis of human renal aging and CKD.

## 3. NAD^+^ Boosting Interventions in Premature Renal Aging and CKD

Different therapeutic strategies are being explored as an approach to increase NAD^+^ levels in age-related diseases to improve healthspan and lifespan. These strategies aim either at stimulating NAD^+^ production, by supplementation with NAD^+^ precursors or stimulation of enzymes involved in NAD^+^ homeostasis, or at limiting NAD^+^ loss by inhibiting its enzymatic consumption or degradation [24]. Augmenting NAD^+^ synthesis with exogenous precursors has been the most widely described strategy in aging and kidney disease and appears to be the most promising approach to date in the context of premature renal aging and CKD. The following sections review the use of NAD^+^ precursor supplementation in preclinical studies and clinical interventions, limited to the context of premature renal aging and CKD.

NAD^+^ precursors are naturally obtained through diet, from a wide variety of our daily foods, such as vegetables, fruits, meat and milk [156]. Firstly, the essential nature of NA, NAM, and NR is emphasized by their general designation as vitamin B3. Although it is commonly accepted that supplementation of all the precursors results in induction of NAD^+^ in most tissues, and that decreased levels interfere with NAD^+^ generation, their distinct roles and efficiencies at physiological concentrations in the kidney are only starting to be understood [24,157]. The precursor of the *de novo* pathway, L-Trp, is also found largely in milk and meat. However, L-Trp is not likely to improve kidney disease as a supplement because of the altered tryptophan metabolism in CKD, which results in the release of uremic toxins [158]. In addition, although the preferred substrate for NAD^+^ biosynthesis in the kidney [28], L-Trp is also a precursor of neurotransmitters and other signaling proteins and is therefore a rather poor precursor of NAD^+^, with a conversion ratio averaging 60:1 in humans [159].

Interestingly, the route of administration of these precursors is critical to their uptake in different tissues and their ability to stimulate NAD^+^. When administered by oral gavage in mice, NA, NAM and NR are directed to the liver and converted to NAD^+^ locally or to NAM, which is released into the circulation and can be assimilated by other organs. In contrast, intravenous administration in mice delivered intact NR and NMN to the kidneys [28]. Furthermore, the *de novo* synthesis and salvage pathways of NAD^+^ in bacteria are much more diverse than in mammals and should not be disregarded in the context of a dietary supplementation [160]. Notably, oral NAM can be deamidated by a microbial nicotinamidase (PncA) in the gut to produce NA that flows into the deamidated salvage pathway, and thus NAM can replenish NAD^+^ levels without requirement for NAMPT activity [161]. Interestingly, the gut microbiota also plays an important role in NR dietary supplementation: after an early phase of incorporation as NR, the precursor is hydrolyzed in the small intestine by CD157 to form NAM, which in turn is deamidated by the microbial PncA [108,161] (Figure 1).

### 3.1. Preclinical Studies

Premature renal aging and CKD can be modeled in rodents in several ways, including mechanical damage to the kidney/urinary system, treatment, or genetic modification (Table 1). Exogenous NAD^+^ precursors have been tested in a variety of models, and the design and results of interventions are described below.

In the context of the Preiss-Handler NAD^+^ metabolism, the efficacy of NA supplementation was studied by Cho et al. in a 5/6 nephrectomy model. At a dose of 50 mg/kg/day in drinking water for 12 weeks, NA reduced proteinuria and hypertension as well as markers of inflammation and oxidative stress such as TGFβ and NF-κB transcription factor activation. A non-statistically significant trend toward decreased serum creatinine was also observed in the NA-treated group [163]. A further study showed an improvement in renal lipid metabolism [164].

Several studies have focused on the NAD^+^ salvage pathways. The effects of NAM were studied in the UUO, aristolochic acid (AA) nephropathy, adenine-induced CKD, and *db/db* models. In the UUO model, NAM injected intraperitoneally (i.p.) at a dose of 200, 400, or 800 mg/kg/day for 14 days, reduced fibrosis, tubular atrophy, apoptosis, cytokine expression (Tumor necrosis factor-alpha (*Tnfα*) and *Il-1β*), and immune cell infiltration [166]. At a dose of 250 mg/kg/day i.p. for 7 days, NAM also reduced fibrosis and levels of neutrophil gelatinase-associated lipocalin (NGAL), a marker of tubular cell injury. NAM also reduced protein acetylation levels, indicating that the beneficial effects of NAM in the UUO model are at least partially due to the enhanced activity of sirtuins [167]. In addition, the same study showed that in a model of AA nephropathy, NAM 500 mg/kg/day i.p. for 15 days reduced tubular injury, NGAL levels, and cytokine expression (*Il-1β* and *Il-6*), but to a lesser extent than in the UUO model [167]. Interestingly, in the adenine-induced CKD model, NAM at 0.3, 0.6, or 1.2% in drinking water successfully prevented the progression of renal dysfunction, inflammation, and tubular injury when given the six weeks of the study but did not improve renal fibrosis and function when taken from week 6 of the adenine regimen to sacrifice at 10 weeks at a dose of 0.3 and 0.6% in drinking water. Moreover, the authors confirmed that the prophylactic administration of NAM increased kidney NAD^+^ and circulating NAM levels. NAM also attenuated the accumulation of glycolysis and Krebs cycle metabolites when administered at an early stage by enhancing NAD^+^-consuming metabolic pathways [173]. Another interesting mechanism of the benefits of NAM on kidney disease is its inhibition of phosphate transport across the small intestine and renal brush borders [179,180], decreasing serum phosphate levels which accumulation leads to mineral and bone disorder (MBD) and eventually cardiovascular disease. In the adenine-CKD rat model, 4 mmol/kg/day i.p. of NAM administered for 6 days significantly decreased serum phosphate and NaPi-2b transporter expression in jejunum brush border membranes, and this was accompanied by less marked elevations in blood urea nitrogen (BUN) and serum creatinine [174]. Finally, in the *db/db* diabetic nephropathy genetic model, NAM was administered for 14 days at 200 mg/kg/day i.p. and no effect was detected on albuminuria or glycated hemoglobin A1c (HbA1c) levels at Week 10 [177]. This study suggests that either NAM may not be the right precursor to administer in early metabolic CKD or that two weeks of treatment is too short to see benefit in this model.

NMN supplementation had positive effects on chronic kidney changes after unilateral IRI in mice, a model to study injury and repair of PTECs upon hypoxic damage, both when administered as a prophylactic medication and during the recovery phase. When administered at 500 mg/kg/day i.p. for the first three days after injury, NMN attenuated tubular senescence, upregulated cytokine expression, and fibrosis at Day 21. When administered at the same dose on days 3 and 14 after surgery, NMN significantly improved NAD^+^ levels in the ischemic kidney and reduced DNA damage, upregulated cytokine expression, and fibrosis at Day 21 [169]. In a diabetic nephropathy setting, Hasegawa et al. reported that renal tubular SIRT1 attenuates albuminuria by increasing NMN levels around glomeruli that in turn epigenetically suppress the overexpression of the tight junction protein Claudin-1 in podocytes, normally expressed only by parietal epithelial cells, and preserve podocyte function. Interestingly, the benefits of PT-specific *Sirt1* overexpression were preserved in two diabetes models, STZ-treated mice and *db/db* mice, but were absent in the 5/6 nephrectomy model [89]. Following this study, they performed a 2-week treatment of *db/db* mice with NMN at 500 mg/kg/day i.p. and confirmed a decrease in albuminuria at Weeks 10 (end of the treatment), 24, and 30, and a dose-dependent decrease at Week 24 with 100–300–500 mg/kg/day NMN. The preventive effect of NMN also ameliorated histological (fibrosis, foot process effacement) and molecular (decreased expression of SIRT1, synaptopodin, NAMPT, and NMNAT1, overexpression of Claudin-1 and decreased NAD^+^ levels) changes in diabetic kidneys [177]. Another study reported an unexpected mechanism by which NMN (120 mg/kg subcutaneously every other day for 20 days) alleviates fibrosis in STZ-induced diabetic rats, proposing that NMN inhibits endogenous NAMPT [145].

While NR and NRH have shown some preclinical potential in AKI [27,29], NR showed no prevention of CKD progression in UUO and POD-ATTAC models and AKI-to-CKD progression after bilateral IRI. UUO and POD-ATTAC mice were pretreated 7 days before CKD induction and then for 7 days and 14 days, respectively, with 800 mg/kg/day of NR in the diet. Although NR successfully replenished NAD^+^ levels in the kidneys, it did not restore renal function or improve interstitial fibrosis [29]. In addition, administered 14 days before bilateral IRI, NR (500 mg/kg/day, i.p.) did not improve renal tubular damage and profibrotic genes expression at Day 14 after IRI [170].

Finally, in the context of diabetic nephropathy, supplementation of STZ-treated spontaneously hypertensive rats with theobromine, a non-polyphenolic component of cocoa, restored NAD^+^ levels and SIRT1 activity. The reduction in NAD^+^ levels is thought to be due to PARP-1 activation and low AMPK activity and correlates with extracellular matrix accumulation and albuminuria [181]. In vitro, NAD^+^ supplementation of glomerular mesangial cells incubated with high glucose confirmed activation of the SIRT1 and AMPK pathways and protected the cells from hypertrophy, an early manifestation of diabetic nephropathy.

Overall, NA, NAM and NMN appear to exert a preventive role against the progression of CKD pathogenesis in preclinical models by attenuating histological and functional damage to the kidney, whereas NR does not consistently demonstrate beneficial effects in the studies cited above, despite increasing renal NAD^+^ levels. Additional use of NR is under investigation in metabolic disease models (*db/db* diabetic mice) which may point at potential benefits in early-stage CKD, although the evidence is still under evaluation [178]. While this wealth of preclinical studies provides overall support to the clinical relevance of targeting NAD^+^ metabolism in CKD, it also raises additional questions. In fact, it highlights the need for further understanding of the differences in renal NAD^+^ metabolism observed in different preclinical models of CKD, and the mechanism of action of NAD^+^ precursors and their functional benefits during CKD settings, especially since the treatment regimen and the model can be very different from one study to another and renal NAD^+^ levels are not systematically measured.

### 3.2. Clinical Interventions

Many clinical trials have been conducted or are underway for CKD treatments, but there is still no reliable therapy for its management. There is growing interest in approaches that include interventions to boost NAD^+^, and several trials have been conducted or initiated to evaluate the benefits of this supplementation in the CKD setting. This section reviews clinical trials using NA and NAM, as for other NAD^+^ boosters, there are only ongoing or initiated trials for CKD (Table 2).

NA supplementation has a long history of success in the treatment of dyslipidemia, being able to reduce triglycerides, total cholesterol and low-density lipoprotein cholesterol (LDL-C) while increasing high-density lipoprotein cholesterol (HDL-C), thereby reducing overall cardiovascular risk by 27% according to a meta-analysis of 30 randomized controlled trials published between 1966 and 2004 [182]. The mechanism underlying these benefits is still incompletely understood but involves the inhibition of free fatty acid mobilization from adipose tissue and both synthesis and catabolism of HDL [183]. Unfortunately, the use of NA in the clinic has been limited by, on the one hand, the subsequent induction of insulin resistance compromising glycemic control in diabetic patients [184], and on the other hand, the painful flushing response [185] since NA acts as an agonist of the G-protein-coupled receptor GPR109A [186] and thus induces the formation of vasodilatory prostanoids [187]. The antagonist of the prostaglandin D2 receptor DP1, laropiprant, has been used in combination with NA to reduce the skin flushes. In addition, the lack of additional effect of NA on cardiovascular disease compared to statins, the first-line LDL-lowering drugs, means that it is no longer recommended for dyslipidemia, except in specific clinical situations [188].

In CKD, NA and NAM have been mainly tested for the treatment of hyperphosphatemia (Table 2). With impaired renal function, phosphate accumulates, and hormonal changes occur in response, so that serum levels of the phosphaturic hormones parathyroid hormone (PTH) and fibroblast growth factor-23 (FGF-23) are markedly increased in CKD. Elevated PTH levels cause bone resorption, and calcium imbalance leads to vascular calcification and endothelial dysfunction. In short, mineral and bone disorder (MBD), the systemic dysregulation of mineral metabolism associated with advanced CKD, is strongly associated with cardiovascular disease (CVD) and poor outcome [189]. NAM is known to inhibit the sodium-dependent phosphate cotransporters NaPi-2a and NaPi-2b located in the renal tubules and small intestine, respectively, thereby increasing its excretion [174,179,180]. NaPi-2b is responsible for more than 90% of the total active phosphate absorption in enterocytes, thus contributing significantly to the maintenance of systemic phosphate homeostasis [190].

Of the studies examining serum phosphate levels, all but one [191] found a decrease after treatment with NA, NAM, or a modified-release formulation of either precursor, ranging from 25 mg/day [192] to 2g/day, and from 8 weeks to 1 year of treatment. However, the study that did not observe a decrease in serum phosphate levels, the COMBINE trial (NCT02258074), had a dropout rate of 25% in the NAM group due to pill burden and gastrointestinal symptoms [191]. In the NICOREN (NCT01011699) and NOPHOS trials, the majority of reported adverse events are also gastrointestinal upset, as well as thrombocytopenia, anemia and pruritus [193,194]. However, a meta-analysis of the efficacy and safety of NAM on phosphate metabolism in hemodialysis patients revealed that thrombocytopenia was the only significant adverse effect caused by NAM [195]. The major metabolite of NAM, N-Me-2PY, also accumulates in serum during NAM treatment and may be partially responsible for the observed side effects [194]. Finally, while most studies have tested NA/NAM in addition to the usual phosphate binders taken by patients, two studies have compared NAM and two different phosphate binders side by side. The COMBINE trial compared NAM with lanthanum carbonate, and both treatments, together and separately, failed to significantly reduce serum phosphate or FGF23 in patients with stage 3–4 CKD over a 1-year period, likely due to limited adherence because of gastrointestinal symptoms [191]. The NICOREN trial (NCT01011699) compared NAM with sevelamer hydrochloride, and the authors concluded that both drugs were equally effective in lowering serum phosphate over 6 months even though the non-inferiority criterion of NAM versus sevelamer was not met due to a high dropout rate [194].

More rarely, the benefits of NA and NAM have been explored on other CKD endpoints rather than MBD markers and lipid metabolism. NA failed to improve flow-mediated dilation, a readout for endothelial cells function, in CKD stages 2 to 4 (NCT00852969). NA, NAM, and niceritrol (an ester of NA used as a prodrug) were mostly unable to ameliorate markers of kidney function such as GFR, BUN, albumin, and creatinine levels [191,196,197,198], except for a small but significant increase in GFR from 53.4 ± 22.0 to 56.6 ± 24.3 mL/min/1.73 m^2^ after 6 months of treatment with 500 mg/day NA in patients with CKD stages 2 to 4 [199]. A study to investigate the impact of NA on proteinuria in patients with diabetic nephropathy was unable to recruit sufficient study subjects (NCT00108485). The NIAC-PKD1 and NIAC-PKD2 studies evaluated the ability of NA to inhibit SIRT1, whose deacetylation activity is linked to cyst growth [200], in patients with autosomal dominant polycystic kidney disease. However, neither the ratio of acetylated p53 protein to total p53 protein in peripheral blood mononuclear cells (PBMCs) nor renal function was altered after one year of treatment with 30 mg/kg/day of NAM [198]. While CKD clinical evidence for NAM still requires further validation, NAM is also currently being tested as a short-term prophylactic drug to prevent AKI after cardiac surgery (NCT04750616, NCT04342975) or septic shock (NCT04589546).

NR was only discovered in 2004 [42], but the fact that it is not vasoactive [201] and therefore does not induce flushing has gained it a lot of interest as an alternative NAD^+^ precursor and it was soon made available as an NR chloride supplement in 2013 under the brand name NIAGEN (Chromadex Inc., Irvine, CA, USA). It was subsequently tested and declared safe in humans at doses up to 2 g/day, and its ability to effectively increase blood NAD^+^ levels was proven [202,203,204,205,206]. In CKD, one study was recently completed in 2021 and another is active, but no clinical data have been disclosed yet. The active study (NCT04040959) aims to evaluate the effects of three months of NR supplementation on arterial stiffness and elevated systolic blood pressure, one of the first potential clinical benefits identified [205]. The CoNR study (NCT03579693) focuses on aerobic capacity, fatigue, mitochondrial function, and cardiac disease in CKD patients without renal replacement therapy after 6 weeks of treatment. Finally, a vitamin B complex administered intravenously for 5 days after an episode of AKI is being tested as a treatment to accelerate recovery of renal function and avoid the transition from AKI to CKD, another phenotype of premature renal aging (NCT04893733).

In light of these mixed results, further clinical research is needed to properly address the alterations of NAD^+^ metabolism in CKD and to evaluate the potential of NAD^+^ exogenous precursors supplementation in terms of clinical benefit, also taking advantage of current active research on NAD^+^ boosters in other clinical settings such as neurodegeneration [207], muscle insulin sensitivity [208], or mitochondrial myopathy [209].

**Table 2 cells-12-00021-t002:** Clinical trials using NAD^+^ precursors in premature renal aging and CKD.

Trial—Year of Completion/Publication	Supplement	Condition	*n* (NAD^+^ Precursor Treated Group)	Primary Endpoint	Secondary Endpoints	Main Results	Ref.
2008	NA 1 g/day, 8 months	CKD on dialysis refractory to treatment of hyperlipidemia and hyperphosphatemia (serum phosphate > 5.5 mg/dL)	9	Serum phosphate levels	Lipids profile, MBD and kidney function markers	↘ Phosphate, TC, TG Unchanged iPTH, LDL-C, albumin, creatinine, BUN	[196]
2010	NA 1 g/day, 4 weeks then 2 g/day, 20 weeks	Dyslipidemia and eGFR < 60 mL/min/1.73 m^2^	110 (with laropiprant), 67 (without laropiprant)	Serum phosphate levels	N/A	↘ Phosphate	[210]
2011	NA 1 g/day, 4 weeks then 2 g/day, 32 weeks	Type 2 diabetes	446	Serum phosphate levels	N/A	↘ Phosphate (No evidence for effect modification by the variable GFR < 60 vs. ≥ 60 mL/min/1.73 m^2^)	[211]
2012	NA 400–1000 mg/day, 8 weeks	Hemodialysis	23	Serum phosphate levels	Lipids profile, MBD markers	↗ HDL-C ↘ Phosphate Unchanged calcium, iPTH, TC, LDL-C, TG	[212]
2012	NA 375–1000 mg/day, 12 weeks	Hemodialysis	14	Serum phosphate levels	Lipids profile, MBD markers	↗ HDL-C ↘ Phosphate Unchanged calcium, iPTH	[213]
NCT00852969—2012	NA 1 g/day vs. NA 100 mg/day, 14 weeks	CKD stages 2–4	30	Flow mediated dilation by brachial artery reactivity (endothelial function)	HDL-C	No significant differences	-
AIM-HIGH NCT00120289—2012	NA 1.5 or 2 g/day + statin, average of 36 months, max 66 months	>45 years old, prevalent CVD and atherogenic dyslipidaemia, sCr < 2.5 mg/L, eGFR_cys+cr_ < 60 mL/min/1.73 m^2^	352	Composite End Point of CHD Death	Other CVD endpoints	No CVD benefit from the addition of NA to statin therapy even if: ↗ HDL-C ↘ TG, LDL-C, phosphate Unchanged iPTH, FGF23, calcium	[188,214]
NCT00108485—2012	NA 1.5–2 g/day, 1 year	Diabetic nephropathy (type II diabetes, CKD stage 2–3, eGFR = 30–89 mL/min/1.73 m^2^, proteinuria < 3.5 g/d, statin treatment)	5	Proteinuria	-	Unable to recruit sufficient study subjects	-
2013	NA 500 mg/day, 6 months	CKD stages 2–4	31	Lipids profile	Serum phosphate levels, GFR	↗ HDL-C, GFR ↘ TG, phosphate	[199]
2014	NA 1 g/day, 4 weeks then 2 g/day, 20 weeks	Dyslipidemia and eGFR = 30–74 mL/min/1.73 m^2^	162 (with laropiprant), 97 (without laropiprant)	Serum FGF23	Other MBD markers	↘ Phosphate, calcium ↘ FGF23, iPTH (for the NA group without laropiprant)	[215]
2016	NA 25–100 mg/day, 3 months	Kidney failure and serum phosphate ≥ 5.5 mg/dL	35	Serum phosphate levels	Lipids profile, MBD markers	↗ HDL-C ↘ Phosphate Unchanged calcium, PTH, TC, TG, LDL-C	[192]
NCT03163576—2022	NA 750 mg/day, 5 weeks then 1.5 g/day, 6 months	Kidney failure, hemodialysis for > 6 months, serum phosphate > 5.0 mg/dL	25	Serum phosphate levels	Lipids profile, MBD markers	↘ Phosphate, TC, LDL-C, TG, iPTH Unchanged HDL-C, calcium	[216]
1998	Niceritrol (ester of NA), 750 mg/day, 8 weeks	Hemodialysis	10	Serum phosphate levels	Kidney functional markers	↗ Serum NA, HDL-C ↘ Phosphate Unchanged BUN, creatinine, calcium, TC, TG	[197]
2004	NAM 500 mg/day, increased by 250 mg/day every 2 weeks until serum phosphate < 6.0 mg/dL (mean dose = 1080 mg/day), 12 weeks total	Hemodialysis and serum phosphate > 6.0 mg/dL	65	Serum phosphate levels	Lipids profile, MBD markers	↗ Serum NAD^+^, HDL-C ↘ Phosphate, iPTH, LDL-C Unchanged calcium, TG	[217]
NCT00508885—2007	NAM 500–1500 mg/day, 8 weeks	Peritoneal dialysis and serum phosphate > 4.9 mg/dL	8	Serum phosphate levels	Lipids profile	↘ Phosphate Unchanged HDL-C, LDL-C, TG	[218]
NCT00316472—2007	NAM 500–1500 mg/day, 8 weeks	Hemodialysis and serum phosphate ≥ 5.0 mg/dL	33	Serum phosphate levels	Lipids profile, MBD markers	↗ HDL-C ↘ Phosphate Unchanged calcium, iPTH, TG, LDL-C, TC	[219]
2008	NAM 500–1000 mg/day, 8 weeks	Hemodialysis and serum phosphate > 5.0 mg/dL	24	Serum phosphate levels	Lipids profile, MBD markers	↗ HDL-C ↘ Phosphate Unchanged calcium, iPTH	[220]
2015	NAM 200–300 mg/day, 6 months	6–18 years old, hemodialysis and serum phosphate > 5.0 mg/dL	30	Serum phosphate levels	Lipids profile, MBD markers	↗ HDL-C ↘ Phosphate, iPTH, TG	[221]
NIAC-PKD1 NCT02140814—2016 & NIAC-PKD2 NCT02558595—2019	NAM 30 mg/kg/day, 12 months	Autosomal dominant polycystic kidney disease	10 & 18	Sirtuin deacetylase activity (acetylated/total p53 ratio in PBMCs)	eGFR, kidney volume, biomarker levels, pain	No sustained inhibition of sirtuin activity Unchanged secondary outcomes	[198]
NICOREN NCT01011699—2017	NAM 0.5–2 g/day, 24 weeks	CKD and hemodialysis	49	Serum phosphate levels	Safety, MBD markers	↘ Phosphate, α-klotho ↗ Calcium, HDL-C, N-Me-2PY Unchanged iPTH, FGF23	[194]
COMBINE NCT02258074—2019	NAM 1.5 g/day, 1 year	CKD stages 3–4 (eGFR = 20–45 mL/min/1.73 m^2^; serum phosphate ≥ 2.8 mg/dL)	51	FGF23, serum phosphate levels	MBD and kidney function markers	↗ Calcium, ACR Unchanged FGF23, phosphate, iPTH, eGFR	[191]
NOPHOS—2020	NAM 250–1500 mg/day, 12 weeks	Hemodialysis and serum phosphate > 4.5 mg/dL	239	Serum phosphate levels	MBD and kidney function markers	↘ Phosphate, iPTH	[193]
CoNR NCT03579693—2021	NR 1.2 g/day, 6 weeks	CKD (eGFR < 50 mL/min/1.73 m^2^) without dialysis or kidney transplantation	25 (*n* total)	Aerobic capacity	PBMCs mitochondrial energetics, fatigue, physical function, heart failure symptoms, oxidative stress, inflammation	Not yet disclosed	-
NCT04040959—Recruiting	NR 1 g/day, 3 months	35–80 years, CKD stages 3–4 (eGFR = 20–60 mL/min/1.73 m^2^), blood pressure < 140/90 mmHg	118 (*n* total)	Carotid-femoral pulse wave velocity	Systolic blood pressure, safety, NAD^+^ metabolome in PBMCs	Active recruitment	-
NCT04893733—Recruiting	Intravenous vitamin B complex, 5 days	AKI	180 (*n* total)	Serum creatinine	Progression to CKD	Active recruitment	-

ACR: Albumin-to-Creatinine Ratio; BUN: Blood Urea Nitrogen; CHD: Coronary Heart Disease; CVD: Cardiovascular Disease; eGFR_cys+cr_: creatinine and cystatin combined equation of estimated Glomerular Filtration Rate; FGF23: Fibroblast Growth Factor 23; HDL-C: High-Density Lipoprotein Cholesterol; iPTH: intact Parathyroid Hormone; LDL-C: Low-Density Lipoprotein Cholesterol; MBD: Mineral and Bone Disorder; PBMCs: peripheral blood mononuclear cells; TC: Total Cholesterol; TG: Triglycerides; ↗: Increased/upregulated; ↘: Decreased/downregulated.

## 4. Conclusions and Future Perspectives

The incidence of premature renal aging and CKD is increasing at an alarming rate and currently there are no effective treatments or biomarkers for early clinical intervention reducing the risk of disease progression. The Kidney Research National Dialogue (KRND) emphasized in 2014 the importance of basic research to target early-stage CKD and slow disease progression [222]. Rather than focusing on the histopathological finding of interstitial fibrosis, it is strongly suggested that research efforts should focus on the proximal tubule and its metabolic responses to injury [14]. For instance, genome-wide transcriptome analysis of healthy human tubules and tubules from CKD patients revealed that metabolism and inflammation are the most dysregulated pathways in CKD. Notably, FAO and its master regulator PGC1α are downregulated in fibrotic kidneys. In mice, transgenic expression of PGC1α in PTECs protected against tubular injury and fibrosis development [19]. NAD^+^ metabolism is one of the major systems controlling energy metabolism and mitochondrial fitness in mammals [40]. Patients with CKD have a decreased ability to synthesize de novo NAD^+^ and increased excretion of its final metabolites, resulting in decreased circulating levels of NAM and NMN, and all correlates with the stage of CKD [22,29]. In addition, several studies have demonstrated accelerated NAD^+^ consumption by PARPs and CD38 in the kidneys of rodents with CKD [106,107,150,153,154,155]. Interventions to restore NAD^+^ levels therefore appear to be a promising approach to target the metabolic perturbation that already occurs in early-stage renal aging and CKD as well as to slow disease progression in later stages.

From a preclinical standpoint, increased NAD^+^ levels via prophylactic NAD^+^ boosting approaches appear to protect against worsening kidney damage in CKD mouse models. From a clinical perspective, the NAD^+^ precursors NA and NAM have been studied primarily as a treatment to reduce serum phosphate levels in more established CKD with functional loss. Although NAD^+^ levels have not been assessed systemically in all interventional studies, the differences observed with the different precursors are intriguing. The fate of these molecules and their mechanism of action, which would explain the potential benefits of NAD^+^ replenishment on metabolic dysregulation in CKD models, need to be further investigated. Importantly, recent evidence of the connection of the amidated and deamidated pathways through base-exchange reactions and the enzymatic activity of the gut microbiota opens new opportunities for dietary supplementation (Figure 1) [108,161]. Taking advantage of the different salvage pathways through a supplementation strategy that bypasses the rate limit of the enzymes and avoids side effects, such as the flushing reaction, would be an effective way to restore renal NAD^+^ levels. Side-by-side comparison of different NAD^+^ precursors, including the recently discovered reduced precursor NRH [27,53], would help to understand their respective potency. Concurrently, the field would benefit from studying other strategies to increase NAD^+^ levels that have shown efficacy in other settings. These strategies include NNMT inhibition [223], NAMPT activation [224,225] although its regulation in CKD needs to be further evaluated, or ACMSD inhibition [226]. In addition, metabolomic and pharmacokinetic studies of the NAD^+^ precursors in CKD patients would clarify the necessary dosage, impact of the gut microbiota, and risk of adverse effects of these compounds in this vulnerable population. The current active growth of the preclinical and clinical space for NAD^+^ boosters will allow for in-depth analysis and informed selection of robust readouts and biomarkers [28,207,208,209].

Finally, CKD is a complex disease that involves multiple differentiated renal cell types and is intimately related in a systemic manner to MBD and CVD [4]. CKD is more generally linked to aging and its cellular and molecular hallmarks such as chronic inflammation, mitochondrial dysfunction, cellular senescence and altered nutrient sensing [227]. Prevention of CKD progression must consider the alterations in body homeostasis that occurs during healthy or premature aging, following renal damage. Recent data show that NAD^+^ stimulation can reduce other hallmarks of aging by supporting mitophagy [228,229], proteostasis [230,231] and preventing cellular senescence [232]. Studies have demonstrated that the inflammatory secretory phenotype of senescent cells can increase CD38 expression in tissues, thereby decreasing NAD^+^ levels [61,114]. Furthermore, senolytics have been found to promote kidney repair and regeneration after injury [233]. Consequently, combining NAD^+^ and senolytic treatments may provide novel therapeutic options. Further studies in this direction, along with deeper characterization of clinical biomarkers for NAD^+^ metabolism alterations and early kidney damage, will be relevant to address translational applications of NAD^+^ boosters in the context of CKD and renal aging.

## Figures and Tables

**Figure 1 cells-12-00021-f001:**
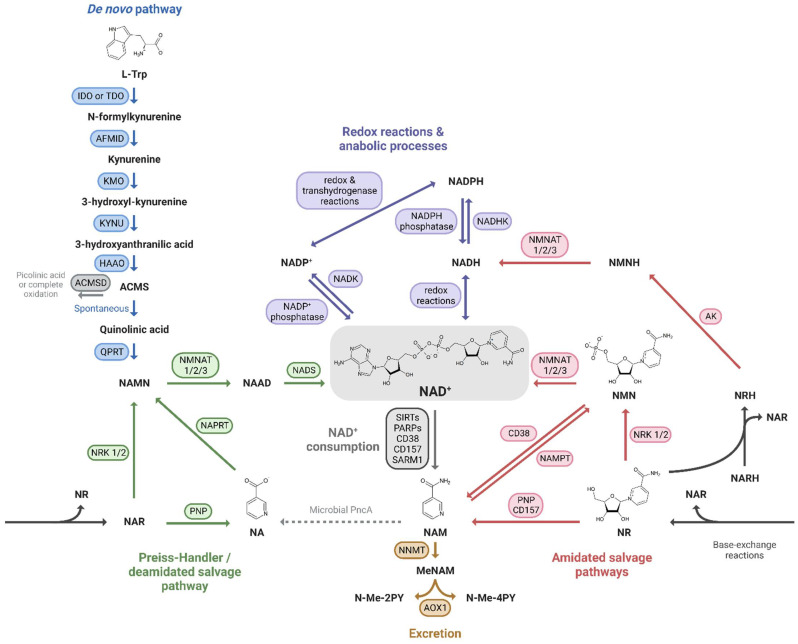
NAD^+^ precursors and biosynthesis pathways. Depiction of the three routes leading to NAD^+^ synthesis: (**i**) the de novo pathway from L-Trp (blue); (**ii**) the Preiss-Handler/deamidated salvage pathway from NA (green); and (**iii**) the amidated salvage pathways including the salvage of NAM to NMN and the conversion of NR and NRH to NMN and NMNH, respectively (red). Intermediate forms of nucleotides in each pathway and reactions linking the amidated and deamidated routes (base-exchange reactions and microbial deamidation) are indicated. The fate of NAD^+^ is also shown: its use in redox reactions, via NADH, and anabolic processes, via the NADP^+^/NADPH couple (purple), its enzymatic consumption (grey), and finally its oxidation and excretion from the body (yellow).

**Figure 2 cells-12-00021-f002:**
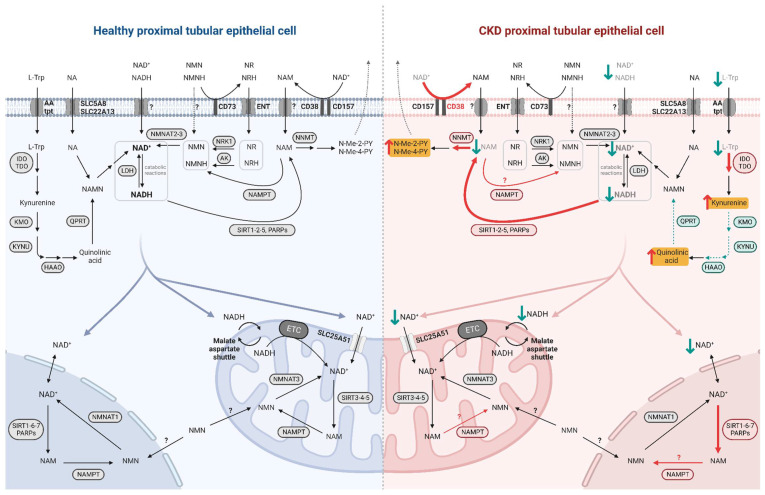
NAD^+^ metabolism in healthy and diseased PTECs. NAD^+^ precursors are taken up by specific transporters at the brush border of PTECs. NAD^+^ biosynthesis occurs primarily in the cytosol while its consumption and salvage can occur in the cytosol, nucleus, and mitochondria, involving isoforms of enzymes specific to these compartments. NAD^+^ moves freely between the nucleus and cytosol while it requires specific transport mechanisms to enter the mitochondria. In diseased PTEC (right side), NAD^+^ biosynthesis is impaired and its consumption is increased, resulting in decreased NAD^+^ levels and increased uremic toxin levels. Enhanced reactions and accumulated metabolites in CKD and premature renal aging are indicated by red arrows, with uremic toxins highlighted in yellow; impaired reactions and decreased metabolites are indicated by green arrows.

**Table 1 cells-12-00021-t001:** Premature renal aging and CKD preclinical models used for NAD^+^ boosting interventions.

Preclinical Model	Description	Phenotype	Interventions & Outcomes			
			NA	NAM	NMN	NR
5/6 nephrectomy (5/6 Nx)	2/3 of a kidney is excised and the second kidney is removed	Model of progressive renal failure after loss of renal mass: persistent decrease in GFR, proteinuria, glomerular sclerosis and hypertension [162]	↘ Hypertension, proteinuria, tubular injury, inflammation markers and lipid accumulation [163,164]	-	-	-
Unilateral Ureter Obstruction (UUO)	Complete ureteral obstruction	Tubulointerstitial fibrosis: interstitial inflammation, tubular dilation, tubular atrophy, macrophage infiltration and extracellular matrix deposition [165]	-	↘ Fibrosis, tubular atrophy, apoptosis, immune cell infiltration, tubular injury and inflammation markers [166,167] ↘ Acetylated levels of Smad 3 and p53 [167]	-	No effect on interstitial fibrosis [29]
Ischemia-Reperfusion Injury (IRI)	Clamping of one or both renal artery and vein for a variable length of time	After 2–9 weeks: AKI-to-CKD transition: renal hypoxia and maladaptive repair leading to interstitial fibrosis [168]	-	-	↘ Fibrosis, PTECs senescence, DNA damage and cytokine expression [169]	↗ Kidney NADH No effect on plasma creatinine, plasma phosphate, tubular damage, profibrotic genes expression [170]
Aristolochic acid (AA) nephropathy	i.p. injection of AA for a variable number of days	After 5–20 days: AKI-to-CKD transition: progressive tubular atrophy and interstitial fibrosis [171]	-	↘ Tubular injury, NGAL levels and cytokine expression (*Il-1β* and *Il-6*) [167]	-	-
Adenine-induced CKD	Adenine supplementation (0.25–0.75%) in the diet for 3–6 weeks	Progressive kidney damage and cardiovascular disease: tubular and glomerular damage, tubulointerstitial fibrosis, inflammation, oxidative stress, loss of renal function, hyperphosphatemia, crystal formation and vessel calcification [172]	-	Prophylactic administration: ↗ Kidney NAD^+^, plasma NAM ↘ Progression of plasma creatinine, BUN and indoxyl sulfate, tubular injury, fibrosis and inflammation markers, accumulation of glycolysis and Krebs cycle metabolites [173] ↘ Serum phosphate, NaPi-2b expression, elevations in BUN and serum creatinine [174]	-	-
POD-ATTAC	Induction of apoptosis in podocytes upon injection of a construct-specific agent	Proteinuria, foot process effacement, mesangial expansion, glomerulosclerosis and tubulointerstitial damage [175]	-	-	-	↗ Kidney NAD^+^ No effect on GFR, plasma BUN and creatinine, and interstitial fibrosis [29]
*db/db*	Deletion mutation in the leptin receptor (*LepRdb/db*)	Type 2 diabetes model: obesity, hyperlipidemia, hyperinsulinemia and insulin resistance Albuminuria, glomerular basement thickening, podocyte loss, oxidative stress and inflammation [176]	-	No effect on albuminuria and HbA1c levels [177]	↗ Kidney NAD^+^, SIRT1, synaptopodin, NAMPT and NMNAT1 expression ↘ Albuminuria, fibrosis, foot process effacement, Claudin-1 expression [177]	↗ Kidney NAD^+^, SIRT3 ↘ Albuminuria, fibrosis, tubular injury, inflammation and oxidative stress markers, and cGAS-STING activation [178]
Streptotozin (STZ)-induced diabetes	i.p. injection of STZ for a variable number of days	Type 1 diabetes model: absolute insulin deficiency Albuminuria, glomerular hypertrophy, glomerular basement thickening, oxidative stress, inflammation and tubulointerstitial fibrosis [176]	-	-	↗ *Sirt1* ↘ *Nampt*, *NF-KB p65*, *Vimentin* [145]	-

↗: Increased/upregulated; ↘: Decreased/downregulated.

## Data Availability

Not applicable.

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
