# Peer review of "NAD+ Metabolism and Interventions in Premature Renal Aging and Chronic Kidney Disease"

_cells, 2022, doi:10.3390/cells12010021_

Round 1

Reviewer 1 Report

In this review, the authors have discussed about the role of NAD+ in chronic kidney disease (CKD). Despite advances in the field of renal biology, no treatment is available to cure CKD till date. Hence, there is an ardent need to develop a treatment. CKD is characterized by low glomerular filtrate and ultimately leads to kidney failure.

Recently several groups have investigated the role of NAD+ in CKD and have found out. This review has beautifully described the role of NAD+ in kidney where they discussed about the possible metabolic route for synthesis of NAD inside the cell and how alteration of NAD+ can lead to the kidney disease. The review also discusses about how boosting NAD+ concentration can intervene with CKD. The most interesting part of the review is the secretion which discusses about the clinical studies which have dissected out the effect of addition of exogenous NAD+ in the murine model.  

Overall, it is a comprehensive review on the NAD+ in kidney disease. The authors have cited all the relevant work in the field and should be of interest to readers involved in studying CKD.

Author Response

We thank the reviewer for the positive evaluation of our manuscript and for pointing out the relevance of summarizing interventional studies, both preclinical and clinical, performed with NAD+ precursors.

Reviewer 2 Report

This manuscript describes the biology of NAD+ in the kidney, and summarized the clinical studies as well. It is a good review paper, and I am sure the readers in the field associated must be benefit from it.

Author Response

We really appreciate the positive feedbacks made by this reviewer, especially about the relevance of the review in the field of kidney aging. Thank you for evaluating our manuscript. 

Reviewer 3 Report

Thank you to the authors for submitting this large and comprehensive review. I have some queries:

1) Please add a label for the figure at the top of page 2

2) Please ensure that your CKD definition conforms to the KDIGO definition (lines 40-42), which is persistent reduction in kidney function or structure for 3 or more months.

3) Please use KDIGO nomenclature (ie, kidney failure rather than ESKD)

4) Please review and revise lines 45-48 which are incorrect in describing CKD.

5) There are minor typos and errors of syntax throughout:

 - line 132

 - line 577

6) please align the style and format of the two tables. Table 1 is difficult to follow because of how it is formatted in the manuscript

7) Please consider a visual abstract to bring all of the many themes and threads together.

Author Response

We thank Reviewer 3 for the valuable comments that helped improving the manuscript.

“1) Please add a label for the figure at the top of page 2”

A label has been added to the figure at the top of page 2, which is the Graphical Abstract.

“2) Please ensure that your CKD definition conforms to the KDIGO definition (lines 40-42), which is persistent reduction in kidney function or structure for 3 or more months”

“3) Please use KDIGO nomenclature (ie, kidney failure rather than ESKD)”

“4) Please review and revise lines 45-48 which are incorrect in describing CKD”

We thank the reviewer for the suggestions on improving the definition of CKD in our manuscript. Based on the points 2-4 above raised by the reviewer, the paragraph describing the definition and classification of CKD (line 43-54) has been revised and rewritten in accordance with the 2012 KDIGO Clinical Practice Guideline for the Evaluation and Management of CKD. Herein we included the CGA classification and adjusted also the nomenclature based on the guidelines.

5) There are minor typos and errors of syntax throughout:

 - line 132

 - line 577”

Thank you for noticing, this has been corrected in the revised manuscript.

“6) please align the style and format of the two tables. Table 1 is difficult to follow because of how it is formatted in the manuscript”

Both tables have been formatted identically in the revised manuscript, the landscape format should make Table 1 more readable. We thank the reviewer again for pointing this out and help improve the readability of our manuscript.

“7) Please consider a visual abstract to bring all of the many themes and threads together.”

We completely agree with the reviewer and understand that this review is large and discusses many results; in fact, we had provided a Graphical Abstract at the top of page 2, which was not clearly labeled before as such, and this has been now corrected following the reviewer’s suggestion at point 1.